# A Critical Review of AMR Risks Arising as a Consequence of Using Biocides and Certain Metals in Food Animal Production

**DOI:** 10.3390/antibiotics12111569

**Published:** 2023-10-27

**Authors:** Christian James, Stephen J. James, Bukola A. Onarinde, Ronald A. Dixon, Nicola Williams

**Affiliations:** 1Formerly Food Refrigeration & Process Engineering Research Centre (FRPERC), Grimsby Institute, Nuns Corner, Grimsby DN34 5BQ, UK; stjames@lincoln.ac.uk; 2National Centre for Food Manufacturing (NCFM), University of Lincoln, South Lincolnshire Food Enterprise Zone, Peppermint Way, Holbeach PE12 7FJ, UK; bonarinde@lincoln.ac.uk; 3School of Life and Environmental Sciences, University of Lincoln, Lincoln LN6 7DL, UK; rdixon@lincoln.ac.uk; 4Institute of Infection, Veterinary and Ecological Sciences, University of Liverpool, Leahurst Campus, Neston CH64 7TE, UK; njwillms@liverpool.ac.uk

**Keywords:** antimicrobial resistance, antimicrobial resistance genes, biocides, farm to fork, food animal production, metals, co-selection

## Abstract

The focus of this review was to assess what evidence exists on whether, and to what extent, the use of biocides (disinfectants and sanitizers) and certain metals (used in feed and other uses) in animal production (both land and aquatic) leads to the development and spread of AMR within the food chain. A comprehensive literature search identified 3434 publications, which after screening were reduced to 154 relevant publications from which some data were extracted to address the focus of the review. The review has shown that there is some evidence that biocides and metals used in food animal production may have an impact on the development of AMR. There is clear evidence that metals used in food animal production will persist, accumulate, and may impact on the development of AMR in primary animal and food production environments for many years. There is less evidence on the persistence and impact of biocides. There is also particularly little, if any, data on the impact of biocides/metal use in aquaculture on AMR. Although it is recognized that AMR from food animal production is a risk to human health there is not sufficient evidence to undertake an assessment of the impact of biocide or metal use on this risk and further focused in-field studies are needed provide the evidence required.

## 1. Introduction

Antimicrobial resistance (AMR) is a complex issue driven by a variety of interconnected factors enabling microorganisms to withstand the killing or static effects of antimicrobial agents, such as antibiotics, antifungals, disinfectants, and preservatives. The widespread use of antimicrobial agents in all contexts is known to result in selection for AMR in microorganisms [1]. There is also evidence that biocidal agents and/or metals may, in some contexts, co-select for AMR in microorganisms.

The focus of this review, commissioned by the UK Food Standards Agency, was to assess what evidence exists on whether, and to what extent, the use of biocides (disinfectants and sanitizers) and certain metals (used in feed and other uses) in animal production (both land and aquatic) leads to the development and spread of AMR within the food chain. A full report of this project is available on the UK Food Standards Agency website [2].

In the context of clinical bacterial infections, resistance is most often defined based on likely clinical efficacy based upon antimicrobial drug/bacteria combinations using recognized standard methods and a clinical breakpoint, which takes into account the pharmacokinetics of the drug. The susceptibility of bacteria to a particular drug is generally assessed by inoculating an isolate into broth or on agar containing different concentrations of the drug to determine the minimum inhibitory concentration (MIC; the lowest concentration of an agent that prevents visible growth of a bacteria) through standardized protocols. The experimentally measured MIC is then compared to standardized clinical breakpoints (discriminatory antimicrobial concentrations used in the interpretation of results of susceptibility testing to define isolates as susceptible, intermediate, or resistant). However, clinical breakpoints are not available for all antimicrobial agents/bacteria combinations. This is particularly the case for agents such as biocides and metals, where no internationally accepted breakpoints exist to define resistance [3,4,5]. While MICs can be determined for biocides and metals, since they are used differently to antibiotics (as discussed in a further section) relying on MIC measurements can be misleading. Furthermore, within the literature “resistance” is not always clearly defined, and reported instances of reduced effectiveness of agents may be at concentrations significantly below the specified in-use concentrations, especially concerning studies investigating susceptibility/tolerance to biocides and metals. For the purposes of this review, the terms tolerance and reduced susceptibility are used when describing biocide and/or metal “resistance”.

### 1.1. Use of Biocides in Food Animal Production

A biocide is defined as an active chemical molecule that controls the growth of, or kills, bacteria and other microorganisms in a biocidal product [3,6,7,8]. Biocidal substances act in different ways and sometimes several biocides are combined within a single product to increase the overall antimicrobial efficacy [8]. The mechanisms of action and tolerance to a wide range of biocides on bacteria have been reviewed and described widely in the literature [5,9,10,11]. Biocides generally act on multiple targets unless present at sub-inhibitory or low concentration [8], although exact mechanisms of action are not fully understood, and are also organism specific. Reported actions include effects on multiple structural and functional components of the bacteria, thereby disrupting cell walls, cell membranes, cross-linking of proteins, and nucleic acids. Reported mechanisms that bacteria use to reduce the impact of biocides include changes in the permeability of the cell membrane to block the biocide; pumps to reduce the intracellular concentration of the biocide; and enzymes to degrade certain biocides. Due to differences in their membrane, Gram-negative bacteria are generally less susceptible to many biocides than Gram-positive bacteria [3].

Biocides, such as Quaternary Ammonium Compounds (QACs), chlorine-releasing agents, and biguanides are widely used in food animal production. Examples of biocide use include: the cleaning and disinfecting of buildings and equipment; decontaminating ponds and equipment in fish farming; in footbaths for operators outside animal housing; in livestock footbaths to treat and prevent the spread of foot infections such as digital dermatitis; to clean the udders of animals used for milk production; and for preserving specific products such as eggs or semen [3,6,8,12]. They may also be used in anti-fouling paints used in aquaculture to reduce the growth of attached organisms on fish cages and nets [13,14]. Biocides are generally not used within body tissues, but some such as organic acids and essential oils (EOs) are added to animal feed and water as antimicrobial controls [3,15].

This review focused on evidence on the impact of biocides used in food animal production on AMR. For this reason, triclosan (5-Chloro-2-(2,4-dichlorophenoxy)phenol) was not considered, as this product was used almost exclusively in human-related products. Due to health concerns and the potential impact on the environment, it has been banned in many countries. While increased AMR to this product is often discussed in the literature, it is of limited relevance to food animal production [16].

### 1.2. Use of Metals in Food Animal Production

Some metals (such as cobalt, copper, iron, manganese, molybdenum, selenium, and zinc) are essential in the diet of living things to maintain various physiological functions and are usually added as nutritional supplements in animal feed [17]. They also have antimicrobial properties and may be used for this purpose in food animal production. The antimicrobial modes/mechanisms of action of metals on bacteria have been reviewed and described in the literature [5,18,19]. As with biocides, the exact mechanisms of action still remain unclear and are also organism specific, but reported actions include effects on the cell wall or membrane; interactions with DNA; binding or inhibition of enzymes and membrane proteins. Reported mechanisms that bacteria use to reduce the impact of metals are similar to those used to reduce the impact of biocides and include enzymes to modify the metal; changes in the permeability of the cell membrane to block the metal; and efflux pumps to reduce the intracellular concentration of the metal. Again, as with biocides, due to differences in their membrane Gram-negative bacteria are generally less susceptible to metals than are Gram-positive bacteria [3].

Copper and zinc are widely used in the pig and poultry sectors as in-feed growth promotors and for enteric disease control [3]. Zinc is also used in aquaculture as a supplement in feed [13,20]. Metals are often used in higher concentrations than needed to ensure adequate nutrition [21,22]. Since the bioavailability of metals in feed is usually quite low, unabsorbed metals are excreted in feces and may accumulate in soil, water, and sediments from food animal production practices. One study in the USA found 90% of in-feed copper and zinc fed to pigs was shed in feces [21]. Although the use of forms of these metals with higher bioavailability (organic forms rather than inorganic) allow for substantial reductions of dietary inclusion rates and consequentially less environmental impact [23,24].

The total amounts and concentrations used of copper and zinc in food animal production may differ among countries, due to restrictions imposed by national legislation. For example, the permitted maximum zinc content in animal feed in the EU (Regulation 2016/1095) is 180 mg zinc/kg for salmonids and in milk replacers for calves; 150 mg zinc/kg for piglets, sows, and all fish species other than salmonids; and 120 mg zinc/kg for other species. The permitted maximum copper content in animal feed in the EU (Regulation 2018/1039) is 15 mg copper/kg for immature bovines (cattle) before the start of rumination; 30 mg copper/kg for other bovines (cattle); 15 mg copper/kg for ovines (sheep); 15 mg copper/kg for caprines (goats); 150 mg copper/kg for piglets suckling and weaned up to 4 weeks after weaning; 100 mg copper/kg for piglets from 5th week after weaning up to 8 weeks after weaning; 50 mg copper/kg for crustaceans; and 25 mg copper/kg for other species.

Other uses of metals include use in livestock footbaths to treat and prevent the spread foot infections such as digital dermatitis [22,25] and wound dressings [3]. Following concerns over therapeutic use of zinc at high concentrations in animal production potentially leading to an increased prevalence of livestock associated methicillin-resistant *Staphylococcus aureus* (LA-MRSA) and environmental contamination, zinc is now only permitted in the EU and UK at concentrations up to 150 ppm for nutritional use [26]. Copper is the principal biocidal component of anti-fouling paints used in aquaculture to reduce the growth of attached organisms on fish cages and nets [13,14]. Copper has also been studied as an antimicrobial alternative to stainless steel surfaces in food production and processing environments [27]. The use of silver and zinc nanoparticles as antimicrobial controls and alternatives to antibiotics in food animal production have received attention in recent years [9,28,29].

### 1.3. Role of Biocides and/or Metals in Co-Selecting AMR

Co-selection mechanisms for biocides and/or metals and clinically as well as veterinary-relevant antibiotics are described widely in the literature [3,5,12,30,31,32], amongst others. There are two main types of related resistance/tolerance co-selection mechanisms:

Cross-resistance—where resistance/tolerance is due to physiological adaptations by the cell that provide similar resistance/tolerance mechanisms to a number of different toxic agents (such as antibiotics, biocides, and metals).

Co-resistance/co-transfer—where resistance/tolerance to different toxic agents is dissimilar but there is a genetic link between resistance/tolerance to different agents, such as the co-location of different resistance genes on the same mobile genetic elements (MGEs), such as plasmids but also on chromosomes. Because of the genetic linkage between such resistance/tolerance, exposure to any of these groups of antimicrobials, or any combination of them, could co-select for the maintenance of the whole MGE and all its associated resistance phenotypes.

Cross-resistance adaptions may be normally present (intrinsic) in the bacteria, or readily acquired by mutation or genetic transfer under appropriate conditions [3,12]. Such adaptions include efflux pumps (transport proteins involved in the extrusion of toxic substrates from within cells into the external environment [33]), biofilm formation, spore formation, nutrient stress responses, and reduced cell envelope permeability [3,12].

Efflux pumps may expel a broad range of unrelated and structurally diverse compounds. Thus, whether intrinsic or acquired, bacteria possessing efflux pumps have substantial potential for cross-resistance to antibiotics, biocides, and/or metals, though this does depend on the nature of the efflux pump [3,33].

Biofilms are complex structures formed by different or single types of bacteria adhering to surfaces which may enhance resistance/tolerance to different antimicrobial agents [34]. Biofilms produce an extracellular matrix that provides a diffusion barrier, plus a potential site for neutralization or binding, of chemical agents, and an enhanced medium for bacterial genetic exchange [3,12]. Bacterial biofilms have been well documented to be highly resistant to antimicrobials [35]. The presence of multiple species of bacteria in biofilms may allow for horizontal gene transfer (HGT) of resistance genes between different bacteria [36]. Biofilms can generate a state of hypermutability (capability for excessive mutation) in bacteria in part due to stress and slower growth that stimulates the development of resistance/tolerance, which may also co-select for AMR [22,34].

Co-resistance/co-transfer may be acquired through the release of resistance genes in MGEs. They may potentially allow some proportion of the bacterial population to survive an otherwise terminal challenge, increasing the risk of selection of organisms permanently adapted to the antimicrobial agent [3]. There can be a genetic link between resistance/tolerance to different agents (co-resistance) through the co-location of different resistance genes on MGEs [3,37,38].

Resistance in many antimicrobial-resistant bacteria (ARB) is encoded by genes that are carried on large conjugative plasmids [39]. These plasmids typically contain multiple antibiotic resistance genes (ARGs) as well as genes that confer reduced susceptibility/tolerance to biocides (BRGs) and/or metals (MRGs), and there are numerous examples reported in the literature [40,41,42,43,44,45]. However, an analysis of the co-occurrence of ARGs, BRGs, and MRGs by Pal et al. [46] concluded that plasmids provide limited opportunities for biocides and metals to promote HGT of AMR through co-selection (though this was more common in bacteria of animal origin), whereas greater possibilities exist for indirect selection (and therefore clonal selection) via chromosomal BRGs and MRGs.

There is evidence that zinc and/or copper may co-select for LA-MRSA due to co-location of the zinc/copper MRG *czrC* and the methicillin resistance gene *mecA* (or its homologue *mecC*) within the staphylococcal cassette chromosome (SCC) *SCCmec* element [47,48,49,50,51,52,53]. *SCCmec* is a MGE that can also transfer to other *Staphylococcus* spp. [54].

There is evidence that some adaptations that enable resistance to antimicrobial agents may result in associated costs to the organism, usually termed “fitness cost”. An example being broad substrate efflux pumps, which consume cell energy resources and indiscriminately remove some useful metabolic substances from the cell [3,16]. Carriage of plasmids (containing resistance genes) have also been cited as another example [39]. However, it has also been reported that compensatory mutations can arise that offset such plasmid fitness costs [55,56].

### 1.4. Role of Concentration of Biocides and/or Metals in Co-Selecting AMR

For the selection of biocide/metal-resistant bacterial strains to occur, some proportion of the population would be expected to survive the application of the biocide/metal. The mode of use of biocides in food animal production would therefore appear to offer fewer opportunities for survivor selection, and consequently co-selection, compared with the use of metals [3,12]. Biocides are generally intended to be lethal/inhibitory, usually after a single application, so are recommended to be used in the field at in-use concentrations that are higher than the MIC determined in the laboratory and to account for different levels of in vitro susceptibility [3,5]. Biocide effectiveness is usually assessed either by time-kill procedures or determination of the concentration that produces a certain log reduction [5]. However, the use of biocides in the presence of heavy organic soiling or with diluting water, which may occur in practice in food animal production, may produce marked reductions in efficacy even at recommended application concentrations and result in sub-inhibitory concentrations being used [3,4]. Furthermore, some biocides (such as organic acids and EOs) may be used in practice at sub-inhibitory concentrations (below MICs) in feed and water as growth promoters and for pathogen control [3].

Low concentrations of antimicrobials in the environment may provide resistant strains of bacteria with a competitive advantage since they may be able to grow in such environments faster than non-resistant strains. The minimum selective concentration (MSC) has been defined as the lowest concentration of an antimicrobial at which resistance/tolerance is positively selected or co-selected. As highlighted by FAO/WHO [57], there are little data on what these threshold MSC values should be to inform suitable standards for biocide and metal concentrations in food animal production. There is evidence that the MSC is affected where species of bacteria are embedded within complex communities, such as animal feces, and may be higher than single-strain-based estimates [58]. FAO/WHO [57] note that the body of evidence to establish such thresholds is likely to take a considerable time to accumulate.

In two reviews, Kampf examined published evidence on the cross-resistance of Gram-positive [59] and Gram-negative [60] bacterial species to biocides. He concluded that there is evidence that sub-inhibitory concentrations of benzalkonium chloride (a QAC used as a sanitizer) and chlorhexidine (a biguanide used as an antiseptic and disinfectant) may co-select for AMR in both Gram-positive and Gram-negative bacteria. There is also evidence for sodium hypochlorite cross-resistance in Gram-negative species, but not in Gram-positives. In contrast he concluded that there is no evidence that cross-resistance to antibiotics has been described after low level exposure to glutaraldehyde, ethanol, propanol, peracetic acid, povidone iodine, and polyhexanide in Gram-positive and Gram-negative bacterial species. Coombs et al. [5] recently came to the same conclusions regarding biocide tolerance and antibiotic cross-resistance. While noting that there was conflicting evidence of antibiotic cross-resistance for chlorine-releasing agents and that while there was no evidence of cross-resistance regarding peracetic acid there was conflicting evidence regarding another peroxygen, hydrogen peroxide.

Unlike biocides, metals are often used at sub-inhibitory concentrations providing more potential for tolerance and co-selection of AMR to emerge [3]. In addition, due to the presence of toxic metals in the general environment, many bacteria have evolved mechanisms of metal tolerance [61]. Yu et al. [22] theorized that certain forms of metals (as stable metal compounds that do not release free metal ions) may provide nutrition to food-producing animals but not be toxic to bacteria, and hence their use in feed would not co-select for AMR. However, there does not appear to be any evidence supporting this hypothesis.

A further mechanism that may be relevant to co-selection is the influence of sub-inhibitory concentrations of biocides and/or metals on gene transfer [3,16]. There is some evidence that while some biocides at sub-inhibitory concentrations may have no effect or inhibit gene transfer [62], some (such as cetrimide, free chlorine, chloramine, and hydrogen peroxide) may increase the efficiency of gene transfer [63,64,65]. Similarly, metals, such as copper and zinc, have also been reported to facilitate HGT of ARGs in water [31,66,67].

It has also been postulated that that low concentrations of antimicrobials may elevate the rate of random mutations in exposed bacterial populations resulting in spontaneous mutants showing cross-resistance to biocides and antibiotics [16,68]. Mutations resulting from biocide exposure have mainly been investigated with triclosan, but some studies have looked at other biocides, such as QACs [35,69].

Even when used at lethal/inhibitory concentrations, it is possible that biocides/metals may contribute to the transmission of AMR by releasing extracellular ARGs (and other resistance genes) from dead bacteria after treatment [70]. Soil and water harbor large environmental reservoirs of a rich microbiota where released genes may integrate into new bacterial genomes through HGT [70,71]. This phenomenon is well-documented in wastewater and drinking water treatment [72] but our literature search did not identify any specific studies that have investigated this in a food animal production context.

## 2. Results

Following the literature search, a total of 3434 titles and abstracts were screened, and 2884 references excluded. A total of 550 publications were considered relevant by title and abstract and full texts collected for second screening. This number was reduced to 148 publications from which some data were extracted, with 402 articles being excluded because they were not in English, the full article was not accessible, or the article was out of scope. A further focused search of Web of Science was carried out prior to submission of this review to ensure any relevant publications released between February and September 2023 were included. This identified a further six publications from which some data were extracted. Overall, there was a total of 154 core publications from which data were extracted.

## 3. Discussion

The literature search identified 22 publications in the last 25 years that in part reviewed aspects of this specific topic [3,6,7,16,22,23,30,31,32,53,57,73,74,75,76,77,78,79,80,81,82,83,84]. None of these reviews cover the entire topic and the majority of these reviews are only focused on land-based food animal production. These reviews have repeatedly highlighted the lack of clear in-field evidence on the role of non-antibiotic drivers in co-selecting AMR in the environment.

Few of these reviews have considered the impact of biocide and/or metal use on AMR in aquaculture. Some reviews mention metals [30,32,75,78,83] and biocides [75] as potential drivers for co-selection of AMR, but they do not cite any specific studies that have addressed the use of metals or biocides used in aquaculture on co-selection of AMR. Our literature search also did not identify any compelling studies that have mapped the use of metals (whether in feeds or in other uses) or biocides in aquaculture with co-selection of AMR.

### 3.1. Impact of Biocides on AMR in Food Animal Production

As highlighted by other reviews [3,6,7,12,31,85], and confirmed in our literature search, while there is much laboratory experimental evidence on the impact of biocides in selecting antibiotic resistance there are considerably fewer field data in relation to the food animal production context. As noted in several reviews, the efficacy of biocidal action in the field and ability to select AMR may be significantly reduced due to the presence of organic soiling or dilution effects. In general, studies do not appear to have specifically quantified these effects on MICs in the context of AMR co-selection. A few studies (as discussed below) have observed that sustained exposure of livestock-associated bacteria to sub-inhibitory concentration of biocides may result in increased levels of AMR among these bacteria.

Studies by Randall et al. [86,87] observed that the use of some biocides commonly used in UK farms can increase bacterial resistance/tolerance to both biocides and antibiotics. A laboratory-based study showed no co-selection effect with ciprofloxacin-resistant strains of *Escherichia coli* to three commercial disinfectants (a tar oil phenol, which was a blend of high boiling point tar acids and organic acid, an oxidizing compound, and a combination of formaldehyde, glutaraldehyde and QAC) [86]. However, there was a slight increase in cyclohexane tolerance among a minority of disinfectant-passaged strains (particularly those subjected to the phenolic biocide). A further laboratory-based study exposed eight *Salmonella* Typhimurium isolates (including field isolates and laboratory mutants) to different farm biocides (a tar oil phenol; an oxidizing compound; an aldehyde-based disinfectant; or QACs) [87]. Results differed depending on the biocide and the *Salmonella* spp. strain tested. Exposure to the aldehyde-based disinfectant reduced susceptibility to the fluoroquinolone ciprofloxacin in some strains. An analysis of proteomes (the complete set of proteins made by an organism) revealed significantly increased expression of the AcrAB–TolC efflux system (responsible for resistance to antimicrobials) after exposure to a tar oil phenol disinfectant. Overall, the results showed that single exposure to biocides was insufficient to select for AMR strains.

Nhung et al. [88] observed that the sustained exposure of *E. coli* and non-typhoidal *Salmonella* (isolated from farmed animals) to sub-inhibitory concentrations of a commonly used commercial disinfectant containing a mix of benzalkonium chloride and glutaraldehyde used on pork and poultry farms in Vietnam appeared to co-select AMR. Increases in MIC for the biocide were strongly correlated with reduced susceptibility shown by increases in MIC (or decreases in inhibition zone) for ampicillin, tetracycline, ciprofloxacin, and chloramphenicol, and to a lesser extent for gentamicin, trimethoprim/sulphamethoxazole. To investigate whether generic efflux pump expression was responsible for the observed changes, the study treated strains with a generic efflux pump inhibitor and measured the changes in AMR before and after treatment. Results suggested that mechanisms other than efflux pumps were responsible for co-selection.

Davies and Wales [16] cited unpublished data from the UK Animal and Plant Health Agency (APHA) that there were concerns that the use of sub-inhibitory concentrations of QACs, because of cost and staff safety issues, was becoming a common practice in UK poultry hatcheries. They reported that there was evidence that certain quinolone-resistant, hatchery-resident *Salmonella* spp. strains appeared to have emerged from such situations and subsequently spread to broilers. This evidence does not appear to have been published elsewhere, or any similar studies undertaken.

The efficacy of cleaning and disinfection regimes in reducing AMR on farms is discussed in reviews such as that by Davies and Wales [16]. It is usually assumed that conventional cleaning and disinfection procedures using biocides that are effective in eliminating susceptible bacteria will be equally effective against antimicrobial-resistant bacteria on farms [16]. However, Davies and Wales [16] cite some unpublished evidence that LA-MRSA may be more resistant than *Salmonella* spp., as well as evidence published by Kotb and Sayed [89] supporting this. A recent study by Montagnim et al. [90] highlighted that multidrug resistant (MDR; defined as resistance to at least one drug within three or more drug classes) strains of *E. coli* may be more resistant to some commonly used farm disinfects than non-MDR strains. A strong association was found between frequent disinfection of pens and colonization of nursery piglets with LA-MRSA in a Canadian study [91]. The study did not, however, map the use of specific biocides to AMR.

Studies have shown some correlation between reduced susceptibility to certain biocides and AMR in bacteria isolated from farmed animals. A German study [92] observed no association of increased didecyl dimethyl ammonium chloride (DDAC, a QAC) MICs with extended-spectrum β-lactamase (ESBL)/AmpC isolates from poultry, but they did observe significant positive correlations for MIC values of DDAC and four antibiotics (chloramphenicol, florfenicol, piperacillin, sulphamethoxazole + trimethoprim) in *E. coli*, as well as for 13 antibiotics in enterococci, suggesting that residual QACs may select antibiotic resistant enterococci. A similar study in China showed an association in reduced susceptibility to sodium hypochlorite and AMR in *Salmonella* spp. isolates from poultry [93]. Positive correlations between chlorine tolerance and clinical antibiotic resistance to ceftiofur, tetracycline, ciprofloxacin and florfenicol were observed. While a study of *E. coli* isolated from pigs, pig carcasses, and pork in Thailand [94] observed some cross-resistance between benzalkonium chloride and chloramphenicol, ciprofloxacin, sulphamethoxazole, and tetracycline; and chlorhexidine and ciprofloxacin, gentamicin, and streptomycin.

The data presented in the literature, however, are conflicting, and numerous other studies have not observed any evidence of cross-resistance or co-selection between biocide use and AMR in bacteria isolated from food animal production environments.

A Brazilian comparison of biocide (sodium hypochlorite and benzalkonium chloride) use on AMR of *S*. Heidelberg isolated from poultry flocks in 2006 with those isolated in 2016 showed no increase in resistance/tolerance over this time period to either biocides or antibiotics (with the exception of tetracycline resistance which showed an increase), suggesting no signs co-selection from biocide use [95].

A survey of strains of *S. enterica* isolated from pigs (132 strains) and poultry (125 strains) in Thailand found that 42% were MDR, but no association with reduced susceptibility to benzalkonium chloride or chlorhexidine was detected in any of the strains [96].

A study of *Salmonella* isolates from two commercial US turkey processing plants found that all *Salmonella* isolates were chlorhexidine tolerant, but no cross-resistance between chlorhexidine and five antibiotics (gentamicin, kanamycin, sulphamethoxazole, streptomycin, and tetracycline) was found in the 130 *Salmonella* spp. serovars compared in the laboratory [97]. A series of later studies by the same US research group [98,99,100] compared biocide tolerance and AMR in strains of *E. coli* O157:H7, *Campylobacter coli*, and *C. jejuni* isolated from cattle, pigs, and poultry, respectively. In all cases no correlation between biocide tolerance and AMR was observed.

A Belgium survey of disinfectant use and resistance in *E. coli* in both poultry and pig production observed no indications for the co-selection of AMR through the use of commonly used biocides (i.e., glutaraldehyde, benzalkonium chloride, formaldehyde, and a formulation of peracetic acid and hydrogen peroxide) in these environments [101]. In a further study by this research team [102] the susceptibility of *E. coli* isolates from a broiler and pig pilot farm to 14 antibiotics and the four disinfectants was monitored over a one-year period. No change in biocide tolerance to these disinfectants was observed and no association was found between biocide use and AMR. In contrast, a German study of *E. coli* from broiler farms did not find a link between phenotypic biocide tolerance to commonly used biocides and AMR on those farms [103]. However, biocide tolerance genes were observed on MGEs in close proximity to ARGs, and some reduced susceptibility to formaldehyde was observed.

As well as their use for cleaning and disinfecting, biocides are used in footbaths and to clean the udders of animals used for milk production [3,6,8,12]. Our literature search identified few studies on the impact of such practices on co-selection for AMR.

Biocides are routinely used in antimicrobial footbaths in commercial dairy farming to prevent lameness caused by bacterial infections [22,25]. A number of different biocides (as well as metals, as discussed in a subsequent section) may be used. During routine cleaning, the contents of these footbaths are usually disposed of into slurry tanks [104]. This may be a potential driver for co-selection of AMR whether through the persistence of biocides within the slurry or through the release of extracellular ARGs from dead bacteria. However, few studies have investigated the impact of footbaths used in food animal production on AMR co-selection or transmission. A study of disinfecting footbaths used in six Norwegian dairy farms found that *Serratia marcescens* may survive and multiply in these baths, but there were no indications of cross-resistance between biocides and AMR in surviving isolates [105].

A number of studies have identified a concern that inappropriate application of teat-dipping biocides applied to dairy cattle could co-select for AMR, although few studies have demonstrated whether this may occur in practice. A study of *Streptococcus uberis* from bovine clinical mastitis in dairy farms with diverse hygienic interventions in Egypt showed that *qac* resistance genes were positively correlated with ARGs/AMR phenotypes in the isolates studied [106]. However, no details were provided on the type of antiseptic used, and no clear evidence of a link between disinfectant use and AMR was demonstrated. A German study [107] observed no cross-resistance in *S. aureus* from cows with subclinical mastitis that showed reduced susceptibility to commercial teat dips (nonoxinol-9 iodine complex and chlorhexidine). An Italian study observed that while 53% of coagulase-negative staphylococci (a cause of subclinical mastitis) isolated from milk showed resistance to at least one of 12 antibiotics tested for and 60% of isolates a reduced susceptibility against benzalkonium chloride and chlorhexidine the isolates had a low prevalence of *qac* genes encoding for disinfectant efflux pumps (12%) and there was no evidence of co-selection [108].

Many essential oils (EOs), plant compounds, and extracts have been shown to act as antimicrobial agents and are promoted as ‘natural’ alternative feed additives to antibiotics in food animal production [15,109,110,111]. While there is much evidence on the efficacy of EOs, there are little data on their modes of action [110] and their potential to drive co-selection of AMR [109]. Since EOs are composed of many chemical constituents, it is not surprising that different oils show synergistic or antagonistic effects to bacteria [112]. De Souza [109] concluded in their review on the effects of sub-inhibitory doses of EOs on AMR that EOs were not likely to impose a major hazard. There is some evidence that controlled exposure of bacteria to sub-inhibitory concentrations of EOs can alter and select for AMR [113], but field studies are lacking. *Thymus maroccanus* (a species of thyme) EO has been shown in vivo to select for AMR in *E. coli* strains [114]. While sub-inhibitory concentrations of tea tree oil (*Melaleuca alternifolia*) have been associated with reduced susceptibility to antibiotics in *E. coli*, *S. aureus*, MRSA, and *Salmonella* spp. [115].

Menthol (an EO) has been suggested as an antibiotic alternative in cattle. US studies [116,117] on feedlot cattle fed menthol (0.3%) reported no increased resistance in *E. coli* isolates to many antibiotics (amoxicillin, ampicillin, azithromycin, cefoxitin, ceftiofur, ceftriaxone, chloramphenicol, ciprofloxacin, gentamicin, kanamycin, nalidixic acid, streptomycin, sulfisoxazole, and sulphamethoxazole); but did observe some reduced sensitivity to tetracycline. It is not clear from these studies why a concentration of 0.3% was used, and trials were not carried out to determine the effect of different concentrations of menthol in feed on co-selection. These studies note that menthol in feed has been shown to promote weight gain in poultry, and there are published studies on its use in fish feed. No other studies appear to have been undertaken to establish whether menthol in feed may select for AMR in other animal species.

Our literature search identified no specific published studies on the effects of the use of biocides on AMR in the aquaculture environment (whether marine or fresh). A study by Romero et al. [118] has been widely cited in the literature as providing evidence of the co-selection of biocides (and metals) on AMR in seafoods. This study observed multiple tolerances/resistances to biocides, metals, and antibiotics in 76% of isolates from a wide range of seafoods purchased at supermarkets and fish markets in the region of Jaen, Spain. ARGs detected included *sul*1 (43.3% of tested isolates), *sul*2 (6.7%), *bla*_TEM_ (16.7%), *bla*_CTX−M_ (16.7%), *bla*_PSE_ (10.0%), *bla*_IMP_ (3.3%), *bla*_NDM−1_ (3.3%), *floR* (16.7%), *aadA1* (20.0%), and *aac(6′)-Ib* (16.7%); and is of concern given that *bla*_IMP_ and *bla*_NDM−1_ encode resistance to carbapenems (CIAs). The only BRG detected was *qacE∆1* (10.0%), but the presence of this BRG suggests that exposure to biocides may co-select for AMR. While many samples were sea-caught fish; sea bass, salmon, and prawn samples were farmed and showed patterns of tolerance/resistance to biocides and antibiotics. However, no direct comparison with any pattern of biocide use during the husbandry of these seafoods was made in this study.

Overall, the literature on investigations of bacterial isolates recovered from the field appear to show some evidence of associations/correlations between certain biocide use and increased resistance to antibiotics. Particularly there is some evidence that QACs, such as benzalkonium chloride, that are widely used in food animal production for disinfection of farm environments and equipment, and chlorhexidine, a biguanide used as an antiseptic and disinfectant for example as a dairy teat disinfectant, may co-select AMR, although there appears to be little clear evidence in the literature for causal links in the field. These biocides have also been identified as risks in other reviews [5,6,59,60]. There is clearly still a need to establish whether current cleaning and disinfection regimes in use in food animal production (both terrestrial and aquatic) represent any real hazard with respect to the selection of AMR.

### 3.2. Impact of Metals on AMR in Food Animal Production

Our literature search identified more published evidence on the impact of metal use in food animal production on AMR than on biocide use. However, as highlighted in other reviews [3,12,31,85], while there is some laboratory experimental evidence on the impact of metals on the selection or development/dissemination of AMR, there are considerably fewer field data (though considerably more than on the impact of biocides). In common with the evidence on the impact of biocides, while there are some data showing an association/correlation in resistance, there is little clear evidence for causal links. Evidence is mainly on the supplementation of pig feed with zinc or copper, or the “therapeutic use” of high concentrations of zinc oxide in pig production. There are little data on the impact of metals on AMR in other forms of food animal production, particularly aquaculture.

As previously mentioned, there is concern that zinc and/or copper may select for LA-MRSA due to the co-location of the MRG *czrC* and the methicillin resistance gene *mecA* (or its homologue *mecC*) within the *SCCmec* element. Other MRGs, including *copB* (encoding copper tolerance), have been found to be present in LA-MRSA and associated with SCC*mec* and integrons [50]. The plasmid pAFS11 obtained from CC398 isolates has been shown to harbor five different ARGs and two MRG operons (including *copA*, encoding copper tolerance) [119]. LA-MRSA strains have been described harboring plasmids carrying MRGs for copper (*copA* and *mco*) and ARGs for multiple antibiotics including macrolides, lincosamides, streptogramin B, tetracyclines, aminoglycosides, and trimethoprim (*erm*(T), tet(L), *aadD*, and *dfrK*) [120]. An association between reduced zinc susceptibility and the development of LA-MRSA CC398 in Danish pigs has been shown to be a consequence of the frequent presence of *czrC* in SCC*mec* (type V) in both pig and human isolates [47,121]. Van Alen et al. [122] reported an increase in the percentage of zinc tolerant LA-MRSA CC398 isolated from patients of a German university hospital located in a pig farming-dense area between 2000 and 2014, which they associated with the use of zinc in pig feed. Prior to 2009, about half of CC398 isolates were zinc tolerant, whereas by 2014 all tested CC398 isolates were found to be zinc tolerant. Zinc tolerance was found to correlate with the presence of the *czrC* gene in all cases. A small-scale study from USA also confirmed a strong association of *S. aureus* CC398 from pigs and the presence of *czrC* [51]. The same study suggests that for certain other lineages (pig-associated LA-MRSA ST5) the contribution of zinc to the emergence of LA-MRSA may be negligible. This may be due to the variations in SCC*mec* cassettes among LA-MRSA lineages and not all will harbor SCC*mec* type V, with ST5 isolates found to carry either SCC*mec* type III or IV, or untypeable cassettes [51]. Argudín et al. [50] demonstrated that the *czrC* gene was almost exclusively found (98%) in the presence of SCC*mec* V in both CC398 and non-CC398 LA-MRSA isolates (CC1 and CC97 LA-MRSA). However, in contrast a Korean study found no evidence of zinc contributing to the prevalence of CC398 and CC5 LA-MRSA strains in pig farms in Korea [123].

Studies have shown that LA-MRSA in weaner pigs is influenced by exposure to therapeutic doses of in-feed zinc (≥2000 ppm) when compared to the recommended dietary concentration (100 ppm). Slifierz et al. [124] reported a significant association between the prevalence of MRSA-positive pigs (followed from birth to weaning) and zinc concentration (3000 vs. 100 ppm) at four and five weeks of age. In both groups, MRSA-positive animals were similarly infrequent by seven weeks of age in the randomized controlled trial. A further report by the same group found a strong association between the concentration of zinc in the nursery ration and colonization of nursery piglets with LA-MRSA [91]. Samples from 390 pigs from 26 farms were compared. Nursery herds testing positive for MRSA reported more frequent use of zinc therapy (≥2000 ppm in-feed), as well as having a higher stocking density. In this study, *czrC* was detected in about two-thirds of isolates in association with a lower susceptibility to zinc compared with *czrC*-negative isolates.

An in vitro study by Peng et al. [125], investigated the growth of two ESBL-producing *E. coli* strains carrying *bla*_CTX-M-1_, with the gene either on a plasmid or chromosomally encoded, in pig fecal material containing an increasing concentration of zinc (0–8 mM). Interestingly, expression of the gene increased with an increasing zinc concentration. The authors suggest that zinc may be inducing the promoter activity of an insertion element (IS) *Ecp1* normally found upstream of the *bla*_CTX-M-1_ gene and increased zinc efflux and thus providing a higher level of zinc tolerance. Furthermore, at higher zinc concentrations there was a higher proportion of CTX-M-1-producing *E. coli* relative to the total flora, but only for the strain where the gene was plasmid encoded. These results suggest that exposure to therapeutic zinc concentrations may give a selective growth advantage to bacteria carrying such plasmid-encoded genes and thereby induce their expression. Such selection does therefore not have to be linked to co-carriage of specific MRGs and ARGs.

Two related studies by Agga et al. [126,127] on the effects of copper supplementation (125 ppm vs. 16.5 ppm) on AMR in weaned pigs used data from the same trial, but analyzed different parameters. The first study [126] found that the MIC for copper was not affected by copper supplementation or by *pcoD* gene carriage (a plasmid-borne copper MRG). The second study [127] reported that copper supplementation was associated with a significant increase in *tetP* genes (which impart resistance to tetracyclines) among fecal *E. coli* but did not show a link with the *pcoD* gene [127]. These studies observed that copper supplementation was associated with lowered *bla*_CMY-2_ gene copies from fecal *E. coli*. According to their results, *tetA* and *bla*_CMY-2_ were positively associated with each other and negatively associated with both *pcoD* and *tetB* genes. They suggested that this points to the potential opportunity to select for a less harmful tetracycline resistance profile in *E. coli* by replacing in-feed antibiotics with copper. A point highlighted in Van Noten et al.’s [76] systematic review of these data. In their review of these data, Wales and Davies [3] concluded that it is possible that baseline levels of antibiotic and copper resistance/tolerance were sufficiently high in this study population that the copper supplementation was insufficient to select for reduced copper susceptibility or associated ARGs. Van Noten et al. [76] judged the trials to be of intermediate methodological quality because of uncertainty concerning the independence of the samples (the same piglet could have been sampled at different weeks). We would agree that the data are not particularly compelling.

A Bavarian study observed that high concentrations of zinc and copper in pig manure (indicative of high concentrations in feed) may promote the spread of AMR of gut microbiota in pigs [128]. In the survey of manure samples from 305 pig farms, the study found that supramedian concentrations of copper (388.5 ppm) and zinc (1199.2 ppm) showed significant associations with *E. coli* phenotypic antibiotic resistance among manure isolates. Bacterial resistance against ampicillin, augmentin (amoxicillin plus clavulanate), and piperacillin was significantly higher in *E. coli* from pig manure containing copper. While resistance rates against piperacillin and doxycycline in *E. coli* from pig manure were associated with zinc.

A possible effect of zinc feed supplementation on the mobility of ARGs in *E. coli* was observed by Bednorz et al. [129], who reported an increased diversity of genotypes and plasmid profiles and increased MDR among weaning pigs supplemented with high concentrations of zinc (>2000 ppm). The study found that 18.6% of the *E. coli* clones from the high zinc group were MDR, but no clones from the control group (50 to 70 ppm). An independent second study [130] used the same feeding setup, but changed the experimental design to focus on a complex analysis of resistance phenotypes rather than clonal diversity. They also observed that high dietary zinc feeding increased the proportion of MDR *E. coli* from weaned piglets, corroborating the finding of the previous study. The impact of zinc was observed in all three habitats tested (feces, digesta, and mucosa). The authors suggested several possible mechanisms for their observations. One was co-selection, as some isolates had both zinc tolerance and antibiotic resistance. Another was enhanced exchange of MGEs under the influence of zinc. Differences in the plasmid profiles of clones of the zinc and control group were observed in the initial study [129].

In contrast, a further study by this group [131] argued against a co-selection mechanism of zinc and AMR suggesting that an explanation for an increase in MDR isolates from piglets with high zinc dietary feeding could be that ARB are more tolerant to stresses such as zinc or copper exposure. In this further study, the group screened the phenotypic zinc/copper tolerance of 210 isolates (including antibiotic resistant, MDR, and non-resistant *E. coli*) selected from two, independent zinc-feeding animal trials. Importantly, no significant association was observed between AMR and phenotypic zinc/copper tolerance of the same isolates.

Medardus et al. [21] also observed an effect of zinc as a feed supplement, and a difference between zinc and copper supplementation. They reported that among 349 *Salmonella* spp. isolates from nine pig units in the USA studied over a two-year period, an elevated zinc MIC was associated with the occurrence of the *czcD*-encoded zinc efflux pump but not with the concentration of fecal zinc. By contrast, fecal copper concentration was associated with an elevated copper MIC, but not with the occurrence of a copper efflux gene *pcoA*. The same study reported that specific serovars were associated both with copper and zinc susceptibility and with patterns of antibiotic resistance; such resistances, however, were not independently associated with copper or zinc susceptibility once serovar was considered. Concentrations of zinc and copper in the feed in these units were between 79–7384 ppm and 3–1384 ppm, respectively.

Studies have reported conflicting evidence on an association between the supplementation of copper in feed of different animals and resistance in fecal enterococci. The development of tolerance to copper in enterococci is associated with the presence of *tcrB*, a copper MRG, which is often located on a conjugative plasmid that may carry ARGs, thus contributing to co-selection [74]. In a study of *Enterococcus faecium* isolated from pigs on Danish farms, *tcrB* was more frequently detected from the more intensively copper-supplemented livestock [132]. Copper tolerance was strongly correlated with macrolide and glycopeptide resistance in isolates from pigs, and *tcrB* genes shown to be located on the same conjugative plasmid as ARGs *erm*(B) and *vanA* (associated with both vancomycin and teicoplanin resistance). In a further study by this group [133], weaner and grower pigs were given a heterogeneous inoculum of *tcrB*-positive and -negative *E. faecium* and reported that exposure at a commercial in-feed concentration of copper (175 ppm vs. 6 ppm) was associated with a higher detection frequency of *tcrB* and of the linked *erm*(B) and *vanA* genes. They also identified the *tcr* genes in the enterococcal species *E. mundtii*, *E. casseliflavus*, and *E. gallinarum*.

In contrast, two US studies by Amachawadi et al. [134,135] found no relationship between feeding weaned piglets with feed with elevated copper concentrations (125 ppm) compared to the control diet (16.5 ppm) and an increased prevalence of copper tolerant enterococci. Though one study by this team [136], similar to the other trials, did show that elevated copper in feed could increase the prevalence of *tcrB*-positive enterococci. These studies did demonstrate a positive correlation between the presence of the *tcrB* gene and tolerance to copper and the possibility of transferring this gene to enterococci from the same and from different species [134,135], but did not specifically examine AMR in these enterococci. A study by Ragland et al. [137] reported no increase in vancomycin-resistant enterococci isolates from piglets (17 to 20 days old) receiving an increased copper or zinc supplementation (192.4 and 2712.7 ppm, respectively) compared to the control group (11.2 and 120 ppm, respectively). In a further study by Amachawadi et al. [138], copper fed to USA feedlot cattle at a growth promotion concentration (100 ppm) was observed to be associated with modest, but significantly increased frequencies (4.5% vs. 2.0% in controls) of detection of *tcrB*-positive and macrolide-resistant *erm*(B)-positive *E. faecium*, whilst resistances to other screened antibiotics, including vancomycin, were unaffected. However, an earlier study reported that feeding elevated copper (up to 100 ppm) and/or zinc (up to 300 ppm) to feedlot cattle had marginal effects on AMR of fecal *E. coli* (resistance to clindamycin, erythromycin, penicillin, tiamulin, tilmicosin, and tylosin) and enterococci (classified as susceptible or intermediate to chloramphenicol, ciprofloxacin, gentamicin, linezolid, penicillin, streptomycin, and vancomycin) [139]. In *E. coli* and *Enterococcus* spp., only minimal differences in MICs of copper, zinc, and antibiotics were noticed. The *tcrB* gene was not detected in feces or in enterococcal isolates. The proportions of *erm*(B) and *tet*(M) were unaffected by copper or zinc supplementation although this was a relatively small-scale trial involving only twenty animals, with only five animals per treatment.

In a study of fecal *E. coli* among 180 weaner pigs in the US, in-feed copper supplementation at a growth-promoting concentration (125 ppm) was associated with reduced susceptibility to chlortetracycline and oxytetracycline in *E. coli* [140]. No significant effects were observed for high concentrations (3000 ppm) of added zinc.

Recent studies of pig fecal samples from a feeding trial carried out in the US in which groups of pigs were fed elevated copper concentrations (250 ppm) in either of two forms, divalent copper sulfate (CuSO_4_) or monovalent copper oxide (Cu_2_O) compared to a control diet (20 ppm), showed no evidence of copper-induced co-selection of ARGs or MGEs known to harbor these genes [141,142]. While recent Portuguese studies of chickens in 7 farms from 2019 to 2020 raised with inorganic and organic copper supplemented feed formulas (below current EU permitted levels), found no difference between the two copper formulas on AMR and no clear evidence of co-selection [143,144]. A high occurrence of MDR, copper-tolerant and colistin-resistant/*mcr*-negative *Klebsiella pneumoniae* was found in the chicken flocks regardless of the feed formulas used and a long-term colistin ban (>2 years) [143]. Neither inorganic or organic copper supplemented feed appeared to selectively promote copper tolerant and MDR *Enterococcus* spp. [144].

A US study that undertook whole-genome sequencing (WGS) of *E. coli* from veal calves found a higher proportion of AMR isolates with BRGs, *sugE* (80%), *sugE1* (27%), and with 50% of isolates carrying the *qacE*∆1 gene [145]. Furthermore, ARGs *mph*(A), *dfrA17*, *aadA5*, and *bla*_CTX-M-15_ were positively associated with silver (*sil*) and copper (*pco*) MRGs. But a negative association was observed between MRGs and some frequently identified ARGs (*sul2, aph*(3”)-Ib, and *aph*(6)-Id). The authors speculated that since copper is found in milk replacer and calf starter diets it may also be co-selecting for AMR, with some association between MGEs and MRGs, BRGs, and ARGs. A Chinese study [146] examining *E. coli* and *Salmonella* spp. isolates from broiler farms and broiler meat found no association between ARGs and MRGs or BRGs in *E. coli*, but there was a positive association between MRGs (including *pcoR* and *zntA*) and BRGs (*sugE*(c), *emrE, mdfA, ydgE/ydgF, qacF, sugE*(p) and *qacE*∆1). In *Salmonella* spp. isolates, ARGs (including ß-lactam resistance genes (*bla*_CTX_, *bla*_TEM_ and *bla*_SHV_), tetracycline resistance genes (*tetA, tetB*, and *tetC*) and sulfonamide resistance genes (*sul1*, *sul2*, and *sul3*) were associated with MRGs (including *pcoR, pcoA*, and *pcoC*), with some MRGs (including *pcoR* and *pcoA*) associated with *qacE*∆1. No details of the *Salmonella* spp. were provided by the authors.

While not evidence of co-selection, a novel genomic island (clusters of genes within a bacterial genome that appear to have been acquired by HGT) likely to be due to the insertion of a plasmid was found in monophasic *S*. Typhimurium isolates from the UK and Italy during 2005–2012 [147]. The genomic island included a number of ARGs genes, but also gene clusters associated with tolerance to zinc and copper. These isolates formed a single clade (isolates composed of a common ancestor) distinct from recent monophasic epidemic clones previously described from North America and Spain. Furthermore, isolates within this clade had a significantly higher MIC for copper sulphate than those outside the clade and without the genomic island. The authors concluded that metal supplements in feed within the gastrointestinal tract of pigs may have contributed to the success of this clade. This is also supported by work in the USA by Medardus et al. [21] who also found a strong association between AMR and metal tolerance among serotypes of *Salmonella* spp. of public health importance.

In two opinions, the EFSA FEEDAP Panel concluded that co-selection in the gut bacteria for tolerance to zinc and copper could not be excluded [148,149]. While the opinion on zinc [148] did not consider its impact on AMR in any detail, the opinion on copper in feed [149] did consider its impact on AMR in detail, which was supported by a systematic literature review by Van Noten et al. [76]. While both of these opinions made recommendations (that were later actioned into EU regulations) for lower permitted concentrations of zinc and copper in animal feeds (as quoted here in a previous section) these concentrations were primarily based on dietary requirements rather than on any impact on AMR co-selection risk. Increasing concern over the therapeutic use of zinc in animal production potentially leading to an increased prevalence of LA-MRSA and environmental contamination with zinc has also contributed to a phase-out of these products in the EU [80]. Therapeutic use of zinc was banned from June 2022 within the EU and was included in UK legislation passed before the UK left the EU. Zinc is now only permitted at concentrations up to 150 ppm for nutritional use, compared to concentrations of 2500 ppm used previously for therapeutic use [26].

As well as in feed, metals may also be used as antimicrobial agents against multiple types of bacteria. Metals, such as zinc and silver, are used for the treatment of burned skin surfaces, open wounds, and specific eye infections and have also been incorporated in medical devices [9,28]. AMR isolates of *E. coli* that also showed reduced susceptibility to silver (and copper) have been isolated from UK pig abattoirs, suggesting that co-selection is possible [150]. Few studies appear to have addressed whether their use as antimicrobial agents could be drivers for co-selection. One of the few published studies to have considered co-selection [151] demonstrated no cross-resistance between silver or gold tolerance (used in the form of nanoparticles) in adapted strains (previously subjected to sub-inhibitory treatments) of *S. aureus* associated with mastitis and isolated from dairy cattle and AMR. Gold nanoparticle treatments were observed to cause less development of resistance than silver treatments. There is evidence that zinc nanoparticles may promote the spread of ARGs and MRGs in soil [152]. In their review of the use of silver as an antimicrobial, Maillard and Hartemann [28] called for a better understanding and control of silver usage to prevent its possible contribution to the spread of AMR. In our opinion, there is still clearly a need to evaluate the potential risk of the use of silver (and other metals) as antimicrobials in food animal production contributing to AMR.

Copper and zinc are routinely used in antimicrobial footbaths in commercial dairy farming to prevent lameness caused by bacterial infections [22,25,104]. Though there appears to be no evidence on their impact on AMR co-selection, it is likely that their disposal into slurry tanks will lead to soil contamination and thus may be a driver for co-selection of AMR [104], but this does not appear to have been studied. Williams et al. [104] estimated that nearly 400 million liters of cattle footbath waste is likely to be disposed annually into slurry tanks in the UK alone. They demonstrated that layered double hydroxides are effective in removing copper and zinc from a commercially available cattle footbath solution and may be a mitigation treatment for reducing this route of contamination.

Few studies on the impact of metals used in aquaculture on AMR were identified in our literature search. While there are studies on the impact of metals in the environment on AMR in fish and seafood, these studies appear to relate mainly to the impact of pollution on wild caught species rather than on the impact of metal use in aquaculture.

Studies have shown a correlation between metal tolerance and AMR in bacteria associated with aquacultural environments. But no direct causative link between the use of metals in feed or as an antifoulant and AMR has been made. For example, a study of *E. coli* from pond sediment from fish farms in Nigeria showed co-occurrence of metal (copper and zinc) tolerance and antibiotic resistance (to β-lactams including 3rd-generation cephalosporins, the fluoroquinolones, potentiated sulphonamides, tetracyclines, aminoglycosides and phenicols) and a significant correlation between concentrations of these metals and AMR [153]. However, there was an absence of detailed information from farms on the use of biocides or feed containing these metals to correlate use with the development of AMR. Similarly, *Aeromonads* and *Pseudomonads* from Australian rainbow trout and sediments displayed resistance to β-lactams, trimethoprim and florfenicol and reduced susceptibility to metals (including zinc and copper) [154]. Again, no link was made to any use of these metals in aquaculture beyond speculation regarding the use of copper to control algae and parasites. Chenia and Jacobs [155] observed a correlation between erythromycin resistance and copper tolerance in bacteria isolated from a South African tilapia aquaculture system, but while the authors suggested that copper use in feed and as an antifoulant could be responsible, this was not specifically investigated. None of these studies investigated the presence of specific ARGs.

A study of the dissemination of resistance genes in duck/fish polyculture ponds, a typical farming model in some parts of China, showed significant correlations between concentrations of copper and zinc and numerous ARGs [156]. Concentrations of copper were significantly and positively correlated with the relative abundance of sul3, *tetT*, *tetW*, *qnrB*, *qnrS*, *fexB*, *sul1*, *sul2*, *tetM*, and *qnrA* genes. With zinc concentrations significantly correlated to relative abundance of *sul2*, *sul3*, *tetM*, *tetA*, *tetT*, *tetW*, *qnrA*, *qnrB*, *qnrS*, *aac*(6′)-Ib, *qepA*, *bla*_SHV_, *cmlA*, *floR*, *fexA*, *cfr*, and *fexB* genes. Again, while the authors suggested that differences in metal levels could be related to different feed formulations, no levels of metals were measured in the feeds used.

As previously mentioned, a study by Romero et al. [118] has been widely cited in the literature as providing evidence of the co-selection of biocides and metals on AMR in seafoods. This study observed multiple tolerances to biocides, metals, and antibiotics in isolates from a wide range of seafoods purchased in the region of Jaen, Spain. The copper MRGs *pcoA*/*copA*, and *pcoR* were detected in 36.7% and 6.7% of selected isolates, respectively. While many samples were sea-caught fish; sea bass, salmon, and prawn samples were farmed and showed patterns of tolerance/resistance to metals and antibiotics. These results suggest that exposure to metals may co-select for AMR, but the study did not carry out any direct comparison with any pattern of the use of metals during the husbandry of these seafoods.

Overall, the literature on bacterial strains recovered from feeding and in-field studies show evidence of associations/correlations between metal use in food animal production and increased resistance/tolerance to antimicrobial agents (Table 1). Particularly, there is evidence that high concentrations of copper or zinc may co-select AMR. This has led to a reduction in permitted concentrations of these metals in recent years in some countries. In our opinion, there is still a need to establish whether current use (in feed and other uses) in food animal production still represents a real hazard with respect to the selection of AMR.

### 3.3. Persistence of Biocides and/or Metals Used in food Animal Production in the Environment

Biocides and/or metals used in food animal production (along with ARB, ARGs, BRGs, and MRGs) may be introduced into soil and water through a number of routes, including direct excretion by the animals, land application of animal manures as fertilizers, irrigation with wastewater, and disposal of antimicrobial treatments (such as footbaths).

The environmental persistence of biocides depends on the nature, action, and use of the biocide. While non-oxidizing biocides (such as QACs) are likely to persist in the environment, oxidizing agents, (such as ozone, hydrogen peroxide, chlorine dioxide, sodium hypochlorite, peracetic acid and iodophors) by their nature are unstable and prone to degradation and rapidly breakdown during use [3,16,157]. While several reviews [3,16,73] express concern regarding the persistence of biocides used in food animal production in the environment, particularly QACs, they cite no specific studies that appear to have studied this or provide evidence of exactly how long biocides used in a food animal production context may persist in the environment. Nor did our literature search identify clear evidence on the fate and persistence of on-farm biocides in-field. A comprehensive review of predicted and measured concentrations and fate of QACs in soils and their implications on AMR development was undertaken by Mulder et al. [158]. They predicted that concentrations of QACs in manure-amended soils could theoretically reach 3.5 ppm after 1 year, assuming zero biodegradation, but highlighted the lack of data on this and whether QACs could accumulate in soil over time.

While many biocides breakdown during use, metals do not biodegrade, are very persistent, and will accumulate in the environment. In England and Wales, food animal production has been estimated to be a major source of environmental contamination by zinc and copper [158,159,160]. Livestock manure was found to be responsible for an estimated 37–40% of total zinc and copper inputs. Denmark has maintained a national monitoring program of metals in the environment for the last 28 years to better understand the effects of these practices on the environment. The values and analyses published in 2016 indicate that the use of pig slurry has led to a significant increase in the measured concentrations of copper and zinc in soil [80]. The persistence of metals in agricultural soil may lead to leaching into natural water, thus impacting on irrigation and aquaculture.

A recent EFSA report highlighted that further research is required to quantify the concentrations of potentially co-selective residues of biocides and metals in manures, agricultural, and aquaculture environments to facilitate risk assessment of the role that they may play in co-selection for AMR [32].

### 3.4. Dissemination of AMR from Animal Manures to Agricultural Soils

Land application of animal manure is a common agricultural practice potentially leading to the dispersal and propagation of ARGs in environmental settings. The dissemination of ARB and ARGs from animal manure and slurry to agricultural soils has been addressed in numerous studies and reviews [161,162]. It was not the purpose of our study to review this evidence, only any specific evidence on the impact of biocides and/or certain metals used in animal production on AMR in this context. There is clear evidence that agricultural soils are a vast reservoir of ARB and ARGs, and that the application of animal manure and/or slurry contributes to this reservoir. Overall, environmental factors can have a high impact on selective pressures, distribution, and diversity of AMR in agricultural soil. Namely, soil characteristics, such as silt, clay, organic matter, and pH, have been shown to correlate with the relative abundance of ARB and ARGs [162]. Different manure sources may influence the fate of resistome in agri-ecosystems with studies demonstrating that the application of pig and poultry manures leading to a greater abundance of ARGs than cattle manure [163,164].

There have been numerous studies and reviews [32,82,162,165,166] on how animal manure may be treated or processed to reduce the transmission of ARB and ARGs into the environment. Commonly used methods include aerobic composting, anaerobic digestion, and aerobic digestion. Other alternative methods include the use of biochar, nano-materials, and bacteriophage (though phages have also been implicated in the transfer of resistance genes within the soil microbiota, albeit experimentally [167]). Ezugworie et al. [166] concluded that no single composting protocol completely eliminated ARGs and that a combination of protocols could yield better results. Available data indicate that none of these methods are effective at eliminating ARB and ARGs. Additional research is needed to determine optimum methods appropriate to the different farming methods that may be used in different countries for reducing/eliminating ARB and ARGs from stored manure prior to use in the environment. A recent EFSA report [32] also highlighted that such measures may increase storage and equipment resources requirements and may reduce the fertilizer value, although the report did not cite specific evidence in relation to this conjecture.

There is some evidence that a delay between the application of manure and plant life cycle (germination, growth, or harvest) of crops may reduce contamination and internalization with ARB and ARGs [32]. Again, according to a recent EFSA BIOHAZ Panel report [32], further research is required to define what a suitable delay may be.

### 3.5. Impact of Biocides Used during Food Animal Production on AMR in Animal Manures and Agricultural Soils

No evidence has been found in the literature on the impact of biocides used in food animal production on ARB or genes detected in manure and manure enriched soils. A 2016 review on the occurrence of biocides in animal manure cited only three studies at the time on the occurrence of biocides in manure [168]. The studies cited were on methods of detection and contained no evidence on what biocides may persist in animal manure. The authors at the time highlighted that studies on the occurrence/persistence of biocides in manure had been neglected, which would appear to still be the case.

### 3.6. Impact of Metals Used during Food Animal Production on AMR in Animal Manures and Agricultural Soils

In the UK, zinc concentrations ranging from <5 to 2500 ppm in manure from commercial farms in England and Wales have been reported (with typical concentrations of approximately 500 ppm) [158]. Data collected in China report that concentrations of these metals are higher in pig manure than other animal manures [169]. This is likely to be the case in other countries, but detailed data are lacking. It is also likely that reductions in the concentrations of metals permitted in food animal production in the EU and other countries may have reduced the concentration of these metals in animal waste, but again data are lacking. The environments close to aquaculture production sites have also been reported to contain elevated concentrations of copper and zinc from fish feed [13]. We have found no clear evidence in the literature linking copper and zinc concentration in fish feed used in aquaculture to levels at production sites, whether in the water or sediment.

Metal concentrations in water, sediment, soil, and manure reported in the literature were compiled by Seiler and Berendonk [30]. They introduced the notion that concentrations of metals may need to accumulate to critical concentration before they can trigger co-selection of AMR. As also highlighted by the FAO and WHO [57] and Arya et al. [170], there are little data on what these threshold values should be in order to inform suitable standards for metal concentrations in food animal production. Arya et al. [170] predicted MSCs of 5.5, 1.6, and 0.15 mg/L for copper, zinc, and silver, respectively. Comparing these thresholds with metal concentrations from slurry and slurry-amended soil from a UK dairy farm that used copper and zinc as additives for feed and in an antimicrobial footbath (at current permitted concentrations) they predicted that the slurry (which contained 22.3 and 32.2 mg/L of copper and zinc, respectively) would be co-selective, but not the slurry-amended soil (which contained only 0.07 and 0.16 mg/L of copper and zinc, respectively).

Studies [164,165,171,172,173,174,175,176,177,178,179,180] have shown a positive correlation between the presence of metals in animal manure and in agricultural soils enriched with animal manure containing metals. The specific source of these metals was not identified. While feed and/or medicines are often cited as probable sources of these metals, studies fail to provide clear evidence of a correlation between concentrations of these metals existing in feed and/or medicines and corresponding levels in manure and manure-enriched soils. Thus, there is no firm evidence of a causative relation in this matter. Microbial communities may be shaped by exposure to different agents and furthermore metals within natural environments have been shown to significantly impact on the structure of microbial communities [181]. There is also some evidence that the presence of metals may have a positive effect on the HGT potential of ARGs in soil [174,182,183,184,185]. How metals enhance the mechanism of genetic transfer is not clear, although there is some evidence that livestock-associated bacteria may carry more MGEs than bacteria from other sources (such as clinical environments) [183]. Manure has been shown to be rich in MGEs carrying MRGs co-occurring with ARGs [180], indicating the importance of MGEs mediating in co-selection. While a recent study by Li et al. [186] suggests that while low levels of copper and zinc in pig manure may alter resistance and MGE compositions, they may not be the primary drive for ARG transmission. It should also be noted that these studies have been carried out in China, where different production regimes may be practiced than in other countries (including the more widespread use of antibiotics). The studies also often lack comparisons with control farms with no use of metals and/or antibiotics, which would allow strong conclusions regarding the use of metals.

Metals may be a continuous pressure on co-selection of AMR during the composting of animal manure and waste bedding [165,166,187,188]. Limits on metal concentrations in compost have been issued by different countries [188]. Biochar (organic material that has been carbonized under high temperatures) [162,165,189] or electro-remediation [17] have been suggested as mitigation treatments for reducing the impact of metals on co-selection of AMR during composting.

### 3.7. Impact of Biocides and/or Metals Used in Animal Production on AMR Transfer from Soil to Crops and Foods of Plant Origin

The dissemination of ARB and ARGs from manure to agricultural soils to crops and foods of plant origin has been addressed in numerous studies and reviews [32,57,161,162,190]. A number of reviews (including recent reviews by FAO/WHO [57] and EFSA BIOHAZ Panel [32]) have cited evidence that ARB and ARGs from manure-amended soils can potentially disseminate from soil microbiota to plant microbiota, and thus may be an important route for AMR transmission in foods of plant origin. Since fruits and vegetables are frequently eaten raw or with minimal processing, they can potentially serve as a source of dietary exposure to ARB and ARGs of animal-origin. It was not the purpose of our study to review this evidence, only the evidence on the impact of biocides and/or certain metals used in animal production on AMR. Bacteria of plant origin have been noted as having an abundance of co-resistance genes [46], but there is little evidence of causal links. Though studies, such as Buta et al. [191], have observed an association between the presence of metals and ARGs in animal manure, which migrated with the manure to agricultural soils enriched with animal manure and hence to crops grown in this soil. Our literature search found that longitudinal studies on the impact of biocides and/or metals during animal production on the transmission of AMR to crops and foods of plant origin are lacking.

### 3.8. Impact of Biocides and/or Metals Used in Animal Production on AMR Transfer to Foods of Animal Origin

While feces, fertilizers of animal origin (for example, manure and slurry), and bedding have been identified as potential transmission routes to the dissemination of ARB and genes in animals and foods of animal origin, there is little information on their importance [32]. Longitudinal studies on the impact of biocides and/or metals during animal production on the transmission of AMR to animals and foods of animal origin are lacking. There is evidence of co-carriage of BRGs/MRGs and ARGs in retail meats [150,192] and high concentrations of zinc increasing the prevalence of LA-MRSA, which is of concern; however, studies showing any clear relationship between the use of biocides and/or metals on the farm contributing to AMR in retail foods of animal origin is lacking.

## 4. Materials and Methods

A systematic review approach was taken to the literature search; however, owing to the paucity of comparable published studies on this topic, a narrative critical review approach was taken to the review of the publications identified. The review question was:

“Do biocides and/or metals used in food animal production have an impact on the development of AMR in the food chain?”

The primary source databases searched were Web of Science, Scopus, and Medline. The searches were restricted to records published from 1990 up to February 2023. The keywords were:

co-selection OR “antimicrobial resistance” OR “antimicrobial resistant” OR “antibiotic resistance” OR “antibiotic resistant” OR “drug resistant” OR “drug resistance” OR “multidrug resistant” OR “multidrug resistance” OR “multi resistance” OR “multi resistant” OR ABR OR AMR OR MDR OR MAR OR AMRG

AND

antiseptic OR biocide* OR disinfectant* OR sanitizer* OR sanitiser* OR “essential oil*” OR “metal*” OR antifouling

AND

“food animal production” OR fish OR seafood OR aquaculture OR salmon OR trout OR cow OR cattle OR dairy OR pig OR swine OR sheep OR lamb OR poultry OR chicken OR turkey OR livestock OR food OR manure OR fertiliser OR feed OR crop* OR “ground water” OR soil OR bedding.

Focused Google and Google Scholar searches were used to identify relevant grey literature.

In total, 2173 citations were initially identified in Web of Science, 2303 in Scopus, and 1404 were identified in Medline. There was some overlap between the databases with 2472 duplicates. An additional 29 records were identified through Google searches, other references, and through contact with authors. For all searches, citations and abstracts were uploaded from each of the electronic databases into Covidence. The following exclusion criteria were applied:(1)The publication contained no relevant data on the impact of biocides and/or metals used in food animal production on the development of AMR;(2)The publication measured irrelevant populations (viruses, fungi, and parasites), interventions (biocide not used in food animal production (for example, healthcare)); used for their surfactant properties, antimicrobial peptides (for instance, bacteriocins); or undesirable metals (such as arsenic (As), cadmium (Cd), mercury (Hg), lead (Pb)), outcomes (did not include impact on ARB or genes).(3)The publication was in a language other than English.

The criteria were independently applied to the abstract of each record by at least two members of the five-member project team. For each citation, a consensus was reached that the citation was relevant for inclusion. Arbitration by a third member of the project team was used to settle conflicting appraisals. Full texts were obtained for all abstracts that passed the inclusion criteria.

## 5. Conclusions

Our literature review (in common with other reviews of this topic) has shown that there is some evidence that biocides and metals used in food animal production may have an impact on the development of AMR, either resulting in reduced susceptibility to drugs or clinically significant resistance. There is clear evidence that metals used in food animal production will persist, accumulate, and may impact on the development of AMR in animal production environments for many years. There is less evidence on the persistence and impact of biocides. There is some evidence that while many biocides will rapidly break-down in the environment, some, for example QACs, may persist. However, there is little evidence on how long this persistence may be in animal production environments, and what the impact on AMR may be. There are also particularly little, if any, data on the impact of biocides/metal use on AMR in aquaculture.

It is widely recognized that AMR in food is a risk to consumer health, and that food animal production has an impact on this risk. While there is certainly a theoretical risk, we have found no published evidence that has specifically demonstrated that the use of biocides and/or metals in food animal production increases the risk of the consumer acquiring clinically significant antibiotic pathogens from food or has quantified that risk. There is no clear evidence on how the use of biocides and/or metals in food animal production may impact on AMR entering the food chain (whether directly through products of animal origin or as a result of crop contamination due to their use in food animal production). The published practical studies that have demonstrated an association between biocide and/or metal use and increased AMR/reduced susceptibility risk in live animals, manure, slurry, or soil, but have not looked at the longitudinal risk to food. It must be noted, however, that there is evidence of the co-carriage of BRGs/MRGs and ARGs in retail meats and bacteria of plant origin.

There does not currently appear to be sufficient evidence to undertake an assessment of risk and further focused in-field studies are needed out to inform this evidence gap and provide the data required to assess this risk. In the meantime, while recognizing the benefits of biocide and metal use in food animal production, as a precautionary measure it would be prudent to develop mitigation measures/strategies that reduce the dissemination of biocides/metals, and ARB and ARGs associated with their use, from food animal production into the environment.

## Figures and Tables

**Table 1 antibiotics-12-01569-t001:** Animal studies that have addressed the impact of the use of metals in feed supplementation or therapeutic use in food animal production on antibiotic resistance (AMR).

Form of Animal Production	Context	Metal *	Bacterial Species	Susceptibility to Antimicrobial Agents	Conclusions	Country	Reference
Pigs	Feed supplementation	Cu	*E. faecium*	Macrolides, Glycopeptide	Association with increased resistance.	Denmark	[132]
Pigs	Feed supplementation	Cu	*E. faecium, E. mundtii, E. casseliflavus, E. gallinarum*	Macrolides, Glycopeptide	Association with increased resistance.	Denmark	[133]
Weaning pigs	Feed supplementation	Cu, Zn	*E. coli*	Chlortetracycline, Neomycin, Oxytetracycline, Tiamulin	Copper associated with increased resistance, but not zinc.	US	[140]
Feedlot cattle	Feed supplementation	Cu, Zn	*E. coli*, *Enterococcus* spp.	*E. coli*: Clindamycin, Erythromycin, Tylosin Penicillin, Tiamulin *Enterococcus*: Chloramphenicol, Ciprofloxacin, Gentamcin, Linezolid, Penicillin, Streptomycin, Vancomycin	Marginal effects on antimicrobial susceptibilities of fecal *E. coli* and enterococci.	US	[139]
Pigs	Therapeutic use	Zn	LA-MRSA	Methicillin, Erythromycin, Penicillin, Tetracycline	Association between zinc resistance gene and methicillin resistance.	Denmark	[47,48]
Weaned pigs	Feed supplementation	Cu	*Enterococcus* spp.	Erythromycin	Association with increased resistance.	US	[136]
Weaning pigs	Feed supplementation	Zn	*E. coli*	Ampicillin, Streptomycin, Chloramphenicol, Gentamicin, Tetracycline, Enrofloxacin, Cefotaxime	Association with increased multi-resistance.	Germany	[129]
Weaned pigs	Feed supplementation	Cu	*E. coli*	-	Association with change in resistance profile (possibly more innocuous).	US	[126,127]
Weaning pigs	Therapeutic use	Zn	LA-MRSA	Methicillin	Association with increased resistance.	Canada	[91,124]
Weaned pigs	Feed supplementation	Cu	*E. faecium, E. faecalis*	-	No association with increased resistance.	US	[134,135]
Cattle	Feed supplementation	Cu	*E. faecium*	Macrolides	Association with increased resistance.	US	[138]
Weaning pigs	Feed supplementation	Zn	*E. coli*	Ampicillin, Streptomycin, Sulfamethoxazole-Trimethoprim, Tetracycline, Enrofloxacin	Association with increased resistance.	Germany	[130]
Weaning pigs	Feed supplementation	Zn	*E. coli*	β-lactamases (Ampicillin or cefotaxime), Tetracyclines (Tetracycline), Aminoglycosides (Streptomycin) and Potentiated Suphonamides (sulphamethoxazole/trimethoprim)	Association with increased resistance, but no evidence of co-selection.	Germany	[131]
Pigs	Feed supplementation	Cu	*E. coli,* gut microbiome	-	No association with increased resistance.	US	[141,142]
Chickens	Feed supplementation	Cu	*K. pneumoniae*, *Enterococcus* spp.	-	Possible association with copper-tolerant and colistin-resistant/mcr-negative *K. pneumoniae*, no evidence of selection of MDR Enterococcus.	Portugal	[142,143]

* Zn = Zinc; Cu = Copper.

## Data Availability

A full report of this project is available on the UK Food Standards Agency website at https://doi.org/10.46756/sci.fsa.ich936.

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
