# Peer review of "A Critical Review of AMR Risks Arising as a Consequence of Using Biocides and Certain Metals in Food Animal Production"

_antibiotics, 2023, doi:10.3390/antibiotics12111569_

Round 1
Reviewer 1 Report
Comments and Suggestions for Authors
The study by James et al. reviews the critical global health concern of antimicrobial resistance (AMR) and its potential connection to the use of biocides and certain metals in animal production. It notes existing evidence suggesting that both biocides and metals in food animal production may contribute to AMR development, with clearer indications for metals compared to biocides. Notably, data gaps, especially in aquaculture, highlight the need for further research. This review emphasizes the urgency of this issue due to its global impact and widespread substance usage, emphasizing the necessity for practical research to assess risks to human health, inform mitigation strategies, and safeguard global health and food systems from the threat of AMR. However, I have some comments to be addressed as follows:
Major comments
- Lines 70 - 75: “This review focused on evidence on the impact of biocides used in food animal production on AMR. For this reason, triclosan (5-chloro-2(2,4-dichlorophenoxy)phenol) was not considered, as this product was used almost exclusively in human related products. Due to health concerns and potential impact on the environment, it has been banned in many countries. While increased AMR to this product is often discussed in the literature, it is of limited relevance to food animal production [14].” – To better address the scope of this review, the authors should also provide some examples of biocides employed in food animal production that have demonstrated an impact on AMR.
- Lines 58 - 60: “The mechanisms of action and resistance/reduced susceptibility to a wide range of biocides on bacteria have been reviewed and described widely in the literature [7-9].” – Please briefly describe the main mechanisms of action and resistance of biocides.
- Lines 80 - 82: “The antimicrobial modes/mechanisms of action of metals on bacteria have been reviewed and described in the literature [16, 17].” - Please briefly describe the main mechanisms of action and resistance of metals.
- Since there are many different evidence in this review, I would recommend providing a Table or Figure to summarize the important evidence regarding the impact of biocides and metals on AMR in bacteria. It provides a visual representation that can enhance the clarity of your findings.
Minor comments
- Lines 76, 117, and 177: “1.1. Use of metals in food animal production”, “1.1. Role of biocides and/or metals in co-selecting AMR”, and “1.2. Role of concentration of biocides and/or metals in co-selecting AMR” – Renumbering the sections should be done, as follows: “1.2. Use of metals in food animal production”, “1.3. Role of biocides and/or metals in co-selecting AMR”, and “1.4. Role of concentration of biocides and/or metals in co-selecting AMR”, respectively.
- Lines 273 - 274: “… to both biocides and to antibiotics” – “… to both biocides and antibiotics”
- Line 275: The full name of “E. coli” should be provided here as first mentioned.
- Line 280: The full name of “S. Typhimurium” should be provided here as first mentioned.
- Lines 318 - 319: “The study, however, did not however map the use of specific biocides to AMR.” – “The study did not, however, map the use of specific biocides to AMR.”
- Line 323: “ESBL/AmpC isolates” could be replaced by “extended-spectrum β-lactamase (ESBL)/AmpC-producing isolates”.
- Lines 338 - 339: “(sodium hypochlorite and benzalkonium chloride [a QAC])” should be replaced by “(sodium hypochlorite and QAC)”.
- Lines 344 - 345: “multiple antibiotic resistance (MAR)” should be replaced by “MDR”.
- Lines 345 - 346: “benzalkonium chloride (a QAC)” should be replaced by “QAC”.
- Line 504: “CTX-M-1 producing E. coli” – “CTX-M-1-producing E. coli”
- Line 577: “tcrB” should be in italic form.
- Line 627: “sugE(c)” should be in italic form.
Author Response
Response to Reviewer 1 comments:
Thank you very much for taking the time to review this manuscript. Please find the detailed responses below and the corresponding revisions/corrections highlighted/in track changes in the re-submitted files.
Major comments
Comment 1: Lines 70 - 75: “This review focused on evidence on the impact of biocides used in food animal production on AMR. For this reason, triclosan (5-chloro-2(2,4-dichlorophenoxy)phenol) was not considered, as this product was used almost exclusively in human related products. Due to health concerns and potential impact on the environment, it has been banned in many countries. While increased AMR to this product is often discussed in the literature, it is of limited relevance to food animal production [14].” – To better address the scope of this review, the authors should also provide some examples of biocides employed in food animal production that have demonstrated an impact on AMR.
Response 1: We agree. Examples of biocides that are used in food animal production have now been provided (Line 81). Examples of biocides employed in food animal production that have demonstrated an impact on AMR are discussed in detail in Section 3.1 (Line 299).
Comment 2: Lines 58 – 60: “The mechanisms of action and resistance/reduced susceptibility to a wide range of biocides on bacteria have been reviewed and described widely in the literature [7-9].” – Please briefly describe the main mechanisms of action and resistance of biocides.
Response 2: We agree. As suggested a short description of the main reported mechanisms of action and tolerance of biocides has been added (Lines 73-80).
Comment 3: Lines 80 - 82: “The antimicrobial modes/mechanisms of action of metals on bacteria have been reviewed and described in the literature [16, 17].” - Please briefly describe the main mechanisms of action and resistance of metals.
Response 3: We agree. As suggested a short description of the main reported mechanisms of action and tolerance of metals has been added (Lines 104-112).
Comment 4: Since there are many different evidence in this review, I would recommend providing a Table or Figure to summarize the important evidence regarding the impact of biocides and metals on AMR in bacteria. It provides a visual representation that can enhance the clarity of your findings.
Response 4: We agree. A table summarising the impact of metals used in feed supplementation or therapeutic use in food animal production has been added (Line 782). Studies on biocide use are far more diverse and difficult to summarise in a table.
Minor comments
Comment 5: Lines 76, 117, and 177: “1.1. Use of metals in food animal production”, “1.1. Role of biocides and/or metals in co-selecting AMR”, and “1.2. Role of concentration of biocides and/or metals in co-selecting AMR” – Renumbering the sections should be done, as follows: “1.2. Use of metals in food animal production”, “1.3. Role of biocides and/or metals in co-selecting AMR”, and “1.4. Role of concentration of biocides and/or metals in co-selecting AMR”, respectively.
Response 5: This has now been corrected.
Comment 6: Lines 273 - 274: “… to both biocides and to antibiotics” – “… to both biocides and antibiotics”
Response 6: This has now been changed as suggested (Line 311).
Comment 7: Line 275: The full name of “E. coli” should be provided here as first mentioned.
Response 7: This has now been corrected (Line 313).
Comment 8: Line 280: The full name of “S. Typhimurium” should be provided here as first mentioned.
Response 8: This has now been corrected (Line 318).
Comment 9: Lines 318 - 319: “The study, however, did not however map the use of specific biocides to AMR.” – “The study did not, however, map the use of specific biocides to AMR.”
Response 9: This has now been corrected (Line 356).
Comment 10: Line 323: “ESBL/AmpC isolates” could be replaced by “extended-spectrum β-lactamase (ESBL)/AmpC-producing isolates”.
Response 10: This has now been corrected (Line 360).
Comment 11: Lines 338 - 339: “(sodium hypochlorite and benzalkonium chloride [a QAC])” should be replaced by “(sodium hypochlorite and QAC)”.
Response 11: This has been changed to “(sodium hypochlorite and benzalkonium chloride)”. Since benzalkonium chloride was mentioned as a QAC earlier in the document (Line 236) further mentions “(a QAC)” every time benzalkonium chloride is mentioned have been removed.
Comment 12: Lines 344 - 345: “multiple antibiotic resistance (MAR)” should be replaced by “MDR”.
Response 12: This has now been corrected (Line 381).
Comment 13: Lines 345 - 346: “benzalkonium chloride (a QAC)” should be replaced by “QAC”.
Response 13: This has been changed to “benzalkonium chloride” since has been mentioned as a QAC earlier in the document (Line 207). We feel that it is important to quote specific QACs.
Comment 13: Line 504: “CTX-M-1 producing E. coli” – “CTX-M-1-producing E. coli”
Response 13: This has now been corrected (Line 540).
Comment 14: Line 577: “tcrB” should be in italic form.
Response 14: This has now been corrected (Line 1039).
Comment 15: Line 627: “sugE(c)” should be in italic form.
Response 15: This has now been corrected.
Reviewer 2 Report
Comments and Suggestions for Authors
Thank you for inviting me to review this manuscript entitled " A critical review of AMR risks arising as a consequence of using biocides and certain metals in food animal production", this review found that there is evidence in the literature that both biocides and metals used in food animal production may have an impact on the development of AMR.
I believe it is a very important review and timely released, however, minor comments were raised and need to be addressed
1. Abstract: must be structured according to the journal guidelines.
2. Transfer the material and methods section after the introduction, according to the journal guidelines.
3. last paragraph of introduction "In the context of clinical bacterial infections.....exist to define resistance" needs elaboration and referencing.
4. in general, this field is very hot and there are many published papers were released in 2023. please add them to the body.
5. clinical implications of the review need elaboration.
6. it would be great if the authors add a table of figure summarizing the review findings.
Author Response
Response to Reviewer 2 comments:
Thank you very much for taking the time to review this manuscript. Please find the detailed responses below and the corresponding revisions/corrections highlighted/in track changes in the re-submitted files.
Comment 1: Abstract: must be structured according to the journal guidelines.
Response 1: The abstract has now been restructured according to the journal guidelines (Line 14).
Comment 2: Transfer the material and methods section after the introduction, according to the journal guidelines.
Response 2: We understand that usually the “Material and methods” section would be expected to follow the “Introduction”, however we have followed the Journals Instructions to Authors (https://www.mdpi.com/journal/antibiotics/instructions) in which this section is between the discussion and conclusions (as we did in a previous review published in this journal).
Comment 3: last paragraph of introduction "In the context of clinical bacterial infections.....exist to define resistance" needs elaboration and referencing.
Response 3: This paragraph has now been elaborated and referenced (Line 45).
Comment 4: in general, this field is very hot and there are many published papers were released in 2023. please add them to the body.
Response 4: We apologise that a typo in the submitted document stated that the literature searches were restricted to records publications from 1990 up to February 2022 rather than February 2023 (Line 954), which was the case giving the understanding that papers released in 2023 had not been reviewed. This was not the case and has been corrected. Nethertheless a further focused search of Web of Science has been carried out to ensure any relevant publications released between February and September 2023 have been included. This has identified a further 6 publications from which some data has been extracted and added to the review. A note of this has also been added to the results section (Line 28).
Comment 5: clinical implications of the review need elaboration.
Response 5: Given the sparce evidence and data gaps highlighted in the review we do not believe that there is sufficient evidence to currently assess the clinically implications. Which has now been made clear in the text (Line 1005).
Comment 6: it would be great if the authors add a table of figure summarizing the review findings.
Response 6: A table summarising the impact of metals used in feed supplementation or therapeutic use in food animal production has been added (Line 782). Studies on biocide use are far more diverse and difficult to summarise in a table. We believe as highlighted in the conclusions that the key reviews findings are that there is not sufficient evidence to undertake an assessment of the impact of biocide or metal use on this risk and further focused in-field studies are needed provide the evidence required.
Reviewer 3 Report
Comments and Suggestions for Authors
The manuscript entitled "A Critical Review of AMR Risks Arising as a Consequence of Using Biocides and Certain Metals in Food Animal Production" highlights the literature review and elaborates on the significance of biocides and heavy metal use in food animal production in relation to the development of AMR.
Although, the topic is interesting and also needs attention by scientist to study the development of AMR in food animals in relation to the use of biocides and metals in feed and sanitation measures, the manuscript is lacking mechanistic review.
I suggest adding or improving the manuscript by providing mechanisms of AMR development or the effect of biocides and metals on the development of AMR to make the manuscript a valuable and important contribution to the field. Furthermore, the tables or figures can be added to improve the manuscript.
Comments on the Quality of English LanguageThe English language is appropriate. Minor checks are required.
Author Response
Reviewer 3 comments:
Thank you very much for taking the time to review this manuscript. Please find the detailed responses below and the corresponding revisions/corrections highlighted/in track changes in the re-submitted files.
Comment 1: Although, the topic is interesting and also needs attention by scientist to study the development of AMR in food animals in relation to the use of biocides and metals in feed and sanitation measures, the manuscript is lacking mechanistic review.
Response 1: It is not clear to the authors whether the reviewer is asking for a review of mechanisms associated with the action of and resistance to biocides and metals, but some text has been added into the text related to this. A further examination of these was beyond the scope of this review and furthermore, is still poorly understood. The work reported was commissioned by the UK food standards agency. The focus of this review was to assess what evidence exists on whether, and to what extent, the use of biocides (disinfectants and sanitizers) and certain metals (used in feed and other uses) in animal production (both land and aquatic) leads to the development and spread of AMR within the food chain. The funder did not commission a mechanistic review.
Comment 2: I suggest adding or improving the manuscript by providing mechanisms of AMR development or the effect of biocides and metals on the development of AMR to make the manuscript a valuable and important contribution to the field.
Response 2: Short descriptions of the main reported mechanisms of action and tolerance of biocides and metals have been added (Lines 73 and 104). The role of biocides and/or metals in co-selecting AMR is discussed in detail in section 1.3.
Comment 3: Furthermore, the tables or figures can be added to improve the manuscript.
Response 3: A table summarising the impact of metals used in feed supplementation or therapeutic use in food animal production has been added (Line 782). Studies on biocide use are far more diverse and difficult to summarise in a table.
Reviewer 4 Report
Comments and Suggestions for Authors
This paper presents clear evidence that biocides and metals used in animal food production may have AMR effects that can lead to reduced drug susceptibility and affect the development of antibiotic resistance in animal production environments. Overall it is well-written review article with clear structure. I only have several minor questions.
My major concern is even natural sourced animal feed, they may also contain non-negligible amount of some metals, is it also contribute to the occurrence of AMR?
Besides, there are several minor issues that should be addressed before the manuscript is considered for publication.
1. Line 71 Please check the spelling of the compound name, consider changing to" 5-Chloro-2-(2,4-dichlorophenoxy)phenol “
2. Line 205, please spell out the full name of QAC upon its first appearance.
Author Response
Reviewer 4 comments:
Thank you very much for taking the time to review this manuscript. Please find the detailed responses below and the corresponding revisions/corrections highlighted/in track changes in the re-submitted files.
Comment 1: My major concern is even natural sourced animal feed, they may also contain non-negligible amount of some metals, is it also contribute to the occurrence of AMR?
Response 1: We agree with the reviewer that this is an interesting point but we do not have data on amounts of metals in natural sourced animal feed or on minimum selective concentrations and in the context of bacterial communities, which may affect this.
Comment 2: Line 71 Please check the spelling of the compound name, consider changing to" 5-Chloro-2-(2,4-dichlorophenoxy)phenol “
Response 2: We have considered and made this change (Line 93).
Comment 3: Line 205, please spell out the full name of QAC upon its first appearance.
Response 3: QACs are now mentioned in the paper earlier and spelled out in full at this point (Line 81).
Reviewer 5 Report
Comments and Suggestions for Authors
Congratulations on this research paper on potential risks in using metals and biocides in aquaculture and agriculture. In particular, the suggestions to look for alternatives to manage and reduce diseases in plants, fish and animals.
Author Response
Response to Reviewer 5 comments:
Thank you very much for taking the time to review this manuscript. We thank the reviewer for their kind and positive comments on our review.
Round 2
Reviewer 1 Report
Comments and Suggestions for Authors
The authors have addressed all of my concerns in this revision.
Reviewer 3 Report
Comments and Suggestions for Authors
Dear Authors,
Thank you for providing a revised and updated version of the manuscript. It is appreciatable that the contents have been improved.
Just check the minor grammatical errors etc.
Comments on the Quality of English LanguageEnglish writing is ok.